# Resected Tumor Outcome and Recurrence (RESTORE) Index for Hepatocellular Carcinoma Recurrence after Resection

**DOI:** 10.3390/cancers15092433

**Published:** 2023-04-24

**Authors:** Daniel Hoffman, Amy Shui, Ryan Gill, Shareef Syed, Neil Mehta

**Affiliations:** 1Department of Surgery, University of California, San Francisco, CA 90095, USA; 2Department of Epidemiology and Biostatistics, University of California, San Francisco, CA 90095, USA; 3Department of Pathology, University of California, San Francisco, CA 90095, USA; 4Division of Transplant Surgery, Department of Surgery, University of California, San Francisco, CA 90095, USA; 5Division of Gastroenterology, Department of Medicine, University of California, San Francisco, CA 90095, USA

**Keywords:** hepatocellular carcinoma, cancer recurrence, liver cancer

## Abstract

**Simple Summary:**

Hepatocellular carcinoma (HCC) is the most common primary liver cancer, and despite best efforts to stratify patients recurrence remains a major issue. Our study attempted to identify what variables are involved in recurrence of HCC after resection and if they be used to stratify an individual patient’s risk of recurrence. We developed a simple-to-implement RESected Tumor Outcome and Recurrence (RESTORE) index comprising three commonly assessed variables: alpha-fetoprotein level, vascular invasion, and tumor burden. The RESTORE index was highly predictive of HCC recurrence risk after resection. The RESTORE index will help identify patients who would potentially benefit from more intensive post-resection surveillance or adjuvant therapeutics.

**Abstract: Importance:**

Although many variables have been associated with increased risk of hepatocellular carcinoma (HCC) recurrence after resection, no simple-to-implement risk score has been developed to determine this post-resection risk. **Objective:** We aimed to identify risk factors for HCC recurrence and develop a risk score for predicting recurrence of HCC in patients who undergo resection with curative intent. **Design:** Single-center retrospective analysis **Setting:** Single-center tertiary care referral hospital (University of San Francisco, California). **Participants:** Patients who underwent resection with curative intent for HCC between January 2005 and May 2019 with complete pathologic findings and recorded follow up. **Main Outcomes and Measures:** Univariate and multivariate Cox regression analysis were used to identify independent risk factors for HCC recurrence. A multivariable Cox proportional-hazard regression model with listwise deletion was used to create a risk score. **Results:** A total of 179 patients were included in the study; 129 (72.9%) were men, and the median (IQR) age was 63 (57–67) years. Median alpha-fetoprotein (AFP) was 12.3 ng/mL at time of resection. Most patients (82%) had a single tumor nodule, and the mean aggregate nodule size was 6.75 cm; 28.4% had evidence of vascular invasion. On multivariable Cox proportional-hazards regression, AFP ≥1000 ng/mL, multinodularity, and vascular invasion were independently associated with HCC recurrence. The RESTORE index was created using stratified pre-operative AFP, vascular invasion, and the presence of a single lesion within or beyond Milan Criteria versus multiple lesions. The RESTORE index ranged from 0–9 (highest patient score was 8) and was highly predictive of HCC recurrence (C statistic 0.70). RESTORE could stratify 5-year post-resection HCC recurrence risk, ranging from less than 25% with a score of 0 to more than 80% with a score of 5–8. **Conclusions and Relevance:** The RESTORE index that we developed and validated is a simple-to-implement and novel risk score for patients undergoing resection for HCC and may help identify those who would benefit most from intensive surveillance strategies or adjuvant therapies.

## 1. Introduction

Hepatocellular carcinoma (HCC) is the most common primary liver cancer and the fourth most common cause of cancer-related death worldwide [1,2,3]. Treatment includes locoregional therapies such as trans-arterial chemoembolization and ablation, as well as curative therapies such as surgical resection and liver transplant [4]. Expanded transplantation criteria and rising HCC incidence have made it a leading indication for transplant listing, although organ availability continues to be a limiting factor [5,6].

Resection remains a cornerstone of treatment for patients with early-stage HCC who are unlikely to gain survival benefit from transplants, and for patients who are not considered for transplants for reasons related both to their tumor and to their medical comorbidities and socioeconomic circumstances. However, resection for HCC also carries a high risk of recurrence with annual rates of ≥10%, and for some patient populations, rates as high has 80% by five years [7,8,9].

The American Joint Commission on Cancer (AJCC) and the Barcelona Clinic Liver Cancer (BCLC) staging systems are the two most widely used staging systems for HCC, and both recommend resection only for early-stage (BCLC-0 and BCLC-A) tumors [10,11]. Although risk factors for recurrence have been identified, including microvascular invasion, alpha-fetoprotein (AFP) levels, and tumor grade, no simple-to-implement risk score is available that could help guide organ allocation away from those patients likely to experience HCC recurrence or to identify patients likely to benefit from intensive surveillance strategies or adjuvant therapies. To address this gap, we sought to develop an easy-to-implement recurrence risk score, the RESected Tumor Outcome and REcurrence (RESTORE) index, for patients undergoing curative resection for HCC.

## 2. Methods

### 2.1. Study Design and Patient Population

This single-center retrospective study was approved by our institutional review board. The study included adult patients (age ≥ 18 years) with pre-operatively diagnosed HCC who underwent resection with curative intent between January 2005 and May 2019. Patients undergoing re-resection for recurrent HCC were excluded.

### 2.2. Data Source and Variables

Data were collected from the electronic medical record and included the following variables: age, sex, race, size, and number of HCC lesions found on pre-operative radiologic examination and on pathological examination of resection specimens, preoperative locoregional therapy, and causes of liver disease.

Recurrence was defined by either radiologically identified recurrence (new LiRADS 5 lesion) or Extrahepatic/LIRAD <5 lesions that lead to the initiation of new therapy (either locoregional, re-resection, transplant, or systemic therapy). Biopsy was routinely performed for recurrence confirmation in the setting of non-LIRAD 5 liver lesions or extrahepatic disease.

Pathology reports of resected livers were reviewed to determine histologic grades based on modified Edmondson criteria, capsular involvement, the presence of vascular invasion, the size and number of viable HCC lesions, R0 vs. R1 margin status, fibrosis, steatosis, and the inflammatory grade of the non-tumor liver. Fibrosis and the inflammatory grade of the non-tumor liver were characterized per the Batts–Ludwig system [12]. Steatosis was characterized per the Brunt system [13].

Previously defined transplant criteria (Milan and UCSF) were used as they are applied to patients with HCC being evaluated for transplant candidacy. Patients were defined as within Milan criteria if they had a single tumor ≤5 cm or three or less tumors ≤3 cm, no evidence of macrovascular invasion, and no evidence of metastasis [1]. Patients were defined as being within UCSF criteria if they had a single tumor ≤6.5 cm or three or less tumors ≤4.5 cm, or a total tumor diameter ≤8 cm [5].

### 2.3. Statistical Analysis and Generation of the RESTORE Index

Univariate and multivariable regression hazard ratios (HRs) for predictors of post-resection HCC recurrence were determined by Cox proportional-hazards regression models and reported with 95% confidence intervals (Cis). Recurrence probabilities were estimated by the Kaplan–Meier method and compared using the log-rank test. Hypothesis tests were two-sided, and the significance threshold was set to 0.05. Predictors of 5-year recurrence, determined using a combination of literature reviews, clinical judgments, and unadjusted analyses, were included in a multivariable Cox proportional-hazards regression model. Listwise deletion was used, as all analysis variables had <5% missing data.

A regression coefficient-based approach was used to develop a points-based scoring system from the selected predictors in the model (Sullivan et al. 2004). Points associated with the presence of a given level of a risk factor were determined by scaling the regression coefficient by the AFP 21–99 category coefficient and rounding to the nearest integer. A Kaplan–Meier cumulative incidence plot stratified by quartiles of the risk score was generated, and differences in these risk strata were assessed using Cox models.

### 2.4. Evaluation of Risk Score Performance

Model performance was evaluated in the validation set using Uno’s c-statistic for survival data [14], along with net reclassification improvement (NRI) and integrated discrimination improvement (IDI) [15,16,17,18,19].

Having previously derived and validated a prognostic scoring system—the RETREAT score—to assess for post-transplant HCC recurrence to help guide management [20,21], we compared its performance to that of the RESTORE index. In brief, the RETREAT score is a prognostic score composed of AFP at the time of liver transplant, the presence or absence of microvascular invasion, and the largest viable tumor diameter plus the number of viable tumors. We also compared the performance of the RESTORE index to that of the Tumor Burden Score (TBS), which is a system initially established for colorectal liver metastases that has been validated in HCC [22,23]. The TBS was defined as in the initial publication, where TBS^2^ = (maximum tumor diameter)^2^ + (number of tumors)^2^. Performance of these scores for predicting 5-year recurrence within Milan criteria was also assessed. The mean c-statistic, NRI, and IDI from 2000 bootstrap replications were reported. A c-statistic of 1 corresponds to perfect discrimination, whereas a value of 0.5 corresponds to no discrimination ability. A c-statistic of 0.7 or higher was considered acceptable. NRI quantifies how well the new risk score reclassifies individuals in terms of estimated risk predictions, as compared to the original RETREAT score. IDI is based on integrated sensitivity and specificity and is equivalent to the difference in discrimination slopes of the two models. Pencina et al. state that the concordance index, NRI, and IDI offer complementary information and recommend reporting all three measures when characterizing the performance of the final model [17].

Statistical analyses were performed using SAS version 9.4 and R version 4.0.2. The “survIDINRI” package in R was used to perform model validation.

## 3. Results

### 3.1. Baseline Characteristics of Study Cohort

A total of 179 patients who underwent HCC resection with curative intent and met inclusion/exclusion criteria were included in the final analytic cohort. Baseline characteristics of the cohort are summarized in Table 1. Median age was 63 (IQR 57–67), most patients were male (72.9%), 41.8% were reported as White, 37.9% Asian, 12.4% Black/African-American, and 7.9% as Other. At the time of resection, 33.5% of patients had HBV, 34.1% had HCV, and 36.3% had cirrhosis. Preoperative locoregional therapy (LRT) had been given in 19.6% of patients. Median pre-operative AFP was 12.3 ng/mL (IQR 3.7, 183.7).

Pre-operative radiological evaluation indicated that 82.1% of patients had a single nodule. The mean aggregate nodule size was 6.47 (95% CI 5.76, 7.17). The mean composite score of the largest viable tumor size and number of viable nodules was 7.8 (95% CI 6.82, 8.77).

When evaluating patients against previously validated transplant criteria (Milan and UCSF) (Table 2), just over half (*n* = 92, 51.4%) of patients were within Milan criteria based on pre-operative imaging, with 14.5% (*n* = 26) of patients falling outside of Milan criteria but within UCSF criteria [5]. These closely paralleled the final pathological assessment, with 48.6% (*n* = 87) of patients within Milan criteria and 12.8% (*n* = 23) of patients outside Milan criteria but within UCSF criteria. Some patients were upstaged on final pathological evaluation, with 34.1% (*n* = 61) of patients outside of UCSF criteria on pre-operative imaging and 38.5% (*n* = 69) on pathology.

Pathological examination showed that 36.3% of the cohort had a Batts–Ludwig fibrosis score of 4, followed by 26.3% with a fibrosis score of 0. Over half (58.1%) of patients had Brunt steatosis scores of 0, followed by 27.4% with a score of 1. Almost a third (30.2%) of patients had a necroinflammation score of 0, with 33.5% and 29.6% having scores of 1 and 2/3 respectively. According to the final pathologic analysis, evidence of vascular invasion was present in 28.1% of patients and evidence of capsular involvement was present 38.5%, but 91% had an R0 resection.

### 3.2. Post-Resection Outcomes

Median post-resection follow-up time was 1312 days (95% CI 1028, 1511). Overall proportion of recurrence was 52%, and median time to recurrence was 615 days (IQR 211-1301). Overall, 63.4% (*n* = 59) of patients who experienced recurrence recurred within 1 year, while, cumulatively, 81.7% (*n* = 76) recurred within 2 years of resection. Most recurrences were intrahepatic (77.4%), whereas the proportion of recurrences within Milan criteria vs. outside Milan criteria were similar (49.5% vs. 50.5% respectively).

### 3.3. Recurrence Prediction

According to Cox proportional-hazard regression, univariate predictors of post-resection recurrence were AFP, number of tumor nodules, largest nodule size, vascular invasion, and capsular involvement (Table 3). There was a significantly increased risk of recurrence with an increasing nodule number, with all patients with three or more nodules experiencing recurrence. For each cm increase in the size of the largest tumor nodule, there was a 3.9% increased risk of recurrence, and for each cm increase in the aggregate nodule size, there was a 5.6% increased risk of recurrence.

In our univariate Cox proportional-hazard model, Asian ethnicity and Batts–Ludwig fibrosis stage 1 (versus fibrosis stage 0) were associated with a lower risk of recurrence. When evaluated with Milan and UCSF criteria [1,5], pre-operative radiologic findings of patients being beyond Milan but within UCSF criteria had an increased risk of recurrence, but those being beyond UCSF criteria did not. On pathological examination, however, being outside Milan criteria and within UCSF, and being beyond UCSF criteria, were significantly associated with increased recurrence risk. Age, gender, etiology of liver disease, pre-operative albumin, pre-operative bilirubin, pre-operative ALBI grade [24], pathological diagnosis of cirrhosis, steatosis, inflammation, tumor grade, and pre-operative (LRT) were not significantly associated with recurrence.

### 3.4. Construction of RESTORE Index and Recurrence Estimation

A multivariable Cox proportional-hazard regression model using listwise deletion was used to create a simplified RESTORE index (Table 4). Compared to patients with pre-operative AFP ≤20, those with AFP ≥100 had nearly twice the risk of 5-year recurrence. Compared to patients without vascular invasion, those with micro/macro-vascular invasion had nearly three times the risk, and compared to those with a single lesion within the Milan criteria, patients with multiple lesions had about 3.4 times the risk.

A Kaplan–Meier cumulative incidence plot stratified by low, medium, and high risk and their associated point values of the risk score (Figure 1) shows that the cumulative incidence of 5-year recurrence had a clearly defined difference in incidence between strata (Stratum 2 vs. 1: HR 2.66, 95% CI 1.25–5.67, *p* = 0.01; Stratum 3 vs. 1: HR 10.42, 95% CI 4.8–22.6, *p* < 0.001).

The classification of pre-operative tumor burdens as within or outside of Milan criteria is an important decision point for directing patients with HCC to resection or transplantation. Patients who were within Milan criteria before resection (as seen on pre-operative radiologic examination) were roughly half as likely to experience recurrence as patients whose resection was beyond Milan criteria (HR 0.52, 95% CI 0.34–0.81, *p* = 0.003).

As shown in Figure 2, patients with the highest RESTORE index (≥5) were more likely to experience HCC recurrence than all others (*p* < 0.001). In terms of pre-operative characteristics, only pre-operative transplant criteria (within Milan criteria vs. beyond Milan and within UCSF criteria vs. beyond UCSF) were associated with a high RESTORE index. The radiological variables—number of nodules, largest nodule, aggregate nodule size, number of nodules + largest nodule, and pre-operative AFP—were not significantly associated with a high RESTORE index.

The RESTORE risk score demonstrated discrimination ability comparable to that of the RETREAT score for overall 5-year recurrence (c = 0.70 vs. 0.69) and for 5-year recurrence within Milan (c = 0.65 vs. 0.64). Neither the RESTORE score nor the RETREAT score had adequate discrimination performance for 5-year Milan recurrence, but RESTORE had acceptable discrimination performance for overall 5-year recurrence. RESTORE produced risk estimates that were at least as accurate as the RETREAT score, but the differences were not significant (NRI 0.11, 95% CI −0.12 to 0.31, *p* = 0.30; IDI 0.00, 95% CI −0.08 to 0.07, *p* = 0.98). However, the RESTORE risk score had better discriminatory ability than the Tumor Burden Score for overall 5-year recurrence (c = 0.70 vs. 0.63).

## 4. Discussion

As the incidence and mortality of HCC has risen in the US and worldwide, the importance of identifying optimal treatment algorithms has greater relevance than ever. Although cancer staging guidelines such as the AJCC and BCLC are key in determining patient prognosis, they may not accurately capture the heterogenous outcomes after HCC resection for patients within the same technical stage [25,26,27]. Ongoing debate surrounds the allocation of livers for transplantation and the extent and efficacy of curative resection [28]. To address the need for better prediction of HCC outcomes, we derived the RESTORE index using three variables highly predictive of HCC recurrence: AFP, microvascular invasion, and the number of tumors (versus single lesions within the Milan criteria) (C statistic 0.70). The RESTORE index was able to clearly stratify 5-year HCC recurrence risk for patients with the lowest score, median score, and highest score. The RESTORE index also had slightly higher discriminatory ability than the RETREAT index in predicting 5-year HCC recurrence. The concordance index of RESTORE was similar to or better than others published on internal validation data [24,29,30]. Additionally, the RESTORE index covers both early and late recurrence periods.

Importantly, the RESTORE index, with its three variables, requires no advanced algorithms to calculate in the clinical setting and can therefore be used by clinicians to help stratify and optimize post-resection surveillance because the patterns of HCC recurrence differ by RESTORE strata with respect to both risk and timing. Most patients in the highest risk strata (score ≥5) who experienced recurrence had that recurrence within 2 years, whereas for patients in the medium risk strata (score 1–4), the predominant period of increased risk was within 3 years (Figure 1). There is some evidence that increased surveillance is associated with increased survival for those patients who experience HCC recurrence [31]. Therefore, one possible surveillance strategy for patients with a high RESTORE risk score (≥5) would be to perform surveillance imaging with the standard modalities (e.g., contrast-enhanced CT or MRI of the abdomen plus AFP) every three months for two years, followed by every six months thereafter. Patients with a moderate RESTORE risk score (1–4) could undergo surveillance every three months for one year, followed by every six months. Finally, patients with the lowest RESTORE score (0) could undergo HCC surveillance every six months. Further studies should address the cost-effectiveness and survival benefit of surveillance for HCC recurrence, specifically within the context of the RESTORE index.

Adjuvant therapy has not been adopted as the standard of care for patients with HCC who undergo resection with curative intent. Some therapies used in the advanced HCC setting, such as sorafenib [32], showed no benefit in the adjuvant setting, but others, such as immunotherapy [33] and radio-immunotherapy [34], are still being evaluated. As additional therapeutics are developed for HCC, the RESTORE index would be a good metric to identify patients who might derive the most benefit from aggressive adjuvant therapy.

Multifocal tumors are a strong predictor of recurrence but are difficult to assess on pre-operative evaluation. In our study, of 32 patients found to have multifocal tumors based on pathologic examination, 14 (43.75%) were initially thought to have unifocal tumors based on pre-operative imaging. Furthermore, given the advances in radiological predictors of microvascular invasion (a key predictor of recurrence), imaging can be used to stratify which patients are candidates for surgery as opposed to other interventions [35]. The early detection of recurrence is important for management of these patients, as some may benefit from potential salvage liver transplants in the setting of recurrence. UNOS allows for biopsy-proven T1 recurrences to be listed without a 6-month delay in many circumstances. At some centers, patients thought to be at extremely high risk of recurrence are evaluated as potential recipients for living donor livers post-operatively, before any chance of recurrence.

Tumor grade has not proven a reliable predictor of recurrence. Other histologic features have been reported, as part of the Recurrence Risk Assessment Score (RRAS), to be more reliable predictors of recurrence [36]. This might serve as a surrogate for vascular invasion on pre-resection biopsy within this proposed scoring system, though additional research is needed. Novel biomarkers such as DCP and AFP-L3% have been shown to correlate with micro-vascular invasion and post-surgical outcomes [37,38,39]. Elevated DCP and AFP-L3% have been associated with high-risk explant pathology and worse survival after liver transplantation [39,40,41,42]. These could be used to further refine the scoring model, especially in the pre-operative setting. Their inclusion in the current study is limited by the lack of uniform availability in our patient population.

There are several limitations to our study. One weakness is that pathological examination of a resection specimen is necessary to exclude (and in most cases to identify) microvascular invasion, precluding its use in the pre-operative setting. Furthermore, as a retrospective cohort study there is uncaptured selection bias from having excluded patients who were deferred from operative intervention at all, or those who underwent liver transplantation. However, important strengths of our study include its ease of implementation and its ability to clearly differentiate patients with high and low risk of recurrence.

## 5. Conclusions

In conclusion, we have developed a novel risk index (RESTORE) that is simple to implement in order to predict a patient’s risk of post-resection HCC recurrence. This index may help improve post-resection surveillance strategies and optimize selection of patients for adjuvant therapies and future trials.

Further work is needed to confirm our study findings, preferably in a multi-center manner. With the advancement of imaging modalities and our ability to assess for microvascular invasion pre-operatively, this index could also be applied and validated in a pre-operative manner.

## Figures and Tables

**Figure 1 cancers-15-02433-f001:**
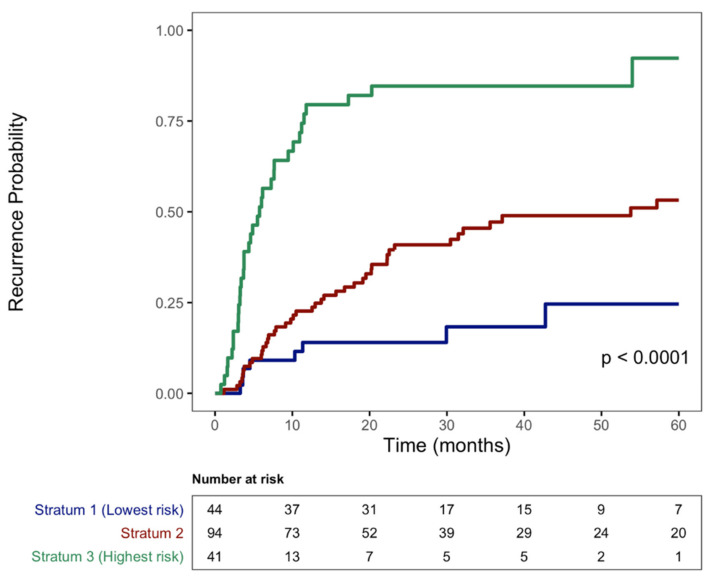
Recurrence by RESTORE index strata (low risk: 0; moderate risk: 1–4, high risk: 5–8).

**Figure 2 cancers-15-02433-f002:**
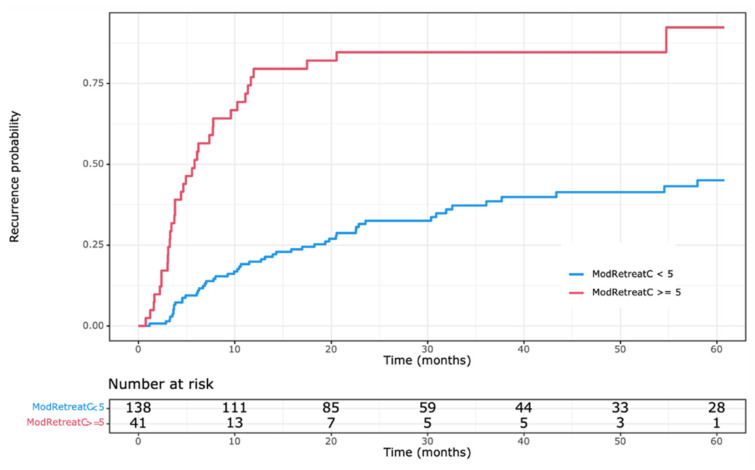
Recurrence by RESTORE index (high risk: 5–8 vs. low/moderate risk: 0–4).

**Table 1 cancers-15-02433-t001:** Demographic and Clinical Characteristics of Patients with HCC.

Variables [*n*, % of Patients Experiencing Outcome]	Overall (*n* = 179)	No Recurrence (*n* = 86)	Recurrence (*n* = 93)
Age, years [median (IQR)]	63 (57–67)	61 (55–67)	63 (57–67)
Age at surgery <50	22 (12.3%)	13 (15.1%)	9 (9.7%)
Age at surgery ≥50	157 (87.7%)	73 (84.9%)	84 (90.3%)
Sex			
Female	50 (27.1%)	27 (31.4%)	23 (24.7%)
Male	129 (72.9%)	59 (68.6%)	70 (75.3%)
Race			
White	74 (41.8%)	26 (30.2%)	48 (51.6%)
Asian	67 (37.9%)	41 (47.7%)	26 (28%)
Black or AA	22 (12.4%)	11 (12.8%)	11 (11.8%)
Other	14 (7.9%)	6 (7%)	8 (8.6%)
Underlying Liver Disease			
HBV	60 (33.5%)	30 (34.9%)	30 (32.3%)
HCV	61 (34.1%)	30 (34.9%)	31 (33.3%)
Cryptogenic	32 (17.9%)	12 (14%)	20 (21.5%)
Unknown/Missing	26 (14.5%)	14 (16.3%)	12 (12.9%)
AFP, ng/mL [median (IQR)]	12.3 (3.7, 183.7)	5.3 (2.6, 59.9)	41 (5.8, 397.5)
≤20	94 (52.5%)	55 (64%)	39 (41.9%)
21–99	27 (15.1%)	12 (14%)	15 (16%)
100–999	29 (16.2%)	9 (10.5%)	20 (21.5%)
1000+	29 (16.2%)	10 (11.6%)	19 (20.4%)
Bilirubin, mg/dL [median (IQR)]	0.80 (0.60, 1.10)	0.80 (0.60, 1.20)	0.80 (0.60, 1)
Albumin, g/dL [median (IQR)]	3.90 (3.30, 4.15)	4.00 (3.42, 4.20)	3.80 (3.10, 4.10)
ALBI Score	−2.57 (−2.82, −2.05)	−2.66 (−2.86, −2.12)	−2.48 (−2.74, −2.02)
ALBI Grade 1	86 (48%)	48 (56%)	38 (41%)
ALBI Grade 2	85 (47%)	33 (38%)	52 (56%)
ALBI Grade 3	8 (4.5%)	5 (5.8%)	3 (3.2%)
Pre-Operative LRT	35 (19.6%)	15 (17.4%)	20 (21.5%)
Pre-Operative TACE	26 (14.5%)	10 (11.6%)	16 (17.2%)
Pre-Operative Y-90	5 (2.8%)	3 (3.5%)	2 (2.2%)
Pre-Operative RFA	2 (1.1%)	1 (1.2%)	1 (1.1%)
Pre-Operative Bland Embolization	2 (1.1%)	1 (1.2%)	1 (1.1%)
**Radiologic Findings**			
Number of Nodules			
1	147 (82.1%)	74 (86.09%)	73 (78.5%)
2	19 (10.6%)	6 (7%)	13 (14%)
3	8 (4.5%)	4 (4.7%)	4 (4.3%)
4+	4 (2.2%)	1 (1.2%)	3 (3.2%)
2+	31 (17.3%)	11 (12.8%)	20 (21.5%)
Largest Nodule Size (cm) (mean, 95% CI)	6.07 (5.41, 6.73)	5.35 (4.36, 6.34)	6.74 (5.85, 7.62)
Aggregate Nodule Size (cm) (mean, 95% CI)	6.44 (5.73, 7.14)	5.54 (4.49, 6.58)	7.25 (6.32, 8.18)
**Liver Pathology**			
Cirrhosis	65 (36.3%)	28 (32.6%)	37 (39.8%)
Fibrosis (missing = 0)			
0	47 (26.3%)	18 (20.9%)	29 (31.2%)
1	17 (9.5%)	14 (16.3%)	3 (3.2%)
2	19 (10.6%)	9 (10.5%)	10 (10.8%)
3	31 (17.3%)	17 (19.8%)	14 (15.1%)
4	65 (36.3%)	28 (32.6%)	37 (39.8%)
Steatosis (missing = 15)			
0	104 (58.1%)	52 (60.5%)	52 (55.9%)
1	49 (27.4%)	25 (29.1%)	24 (25.8%)
2/3	11 (6.1%)	8 (9.3%)	3 (3.2%)
Inflammation (missing = 12)			
0	54 (30.2%)	27 (31.4%)	27 (29%)
1	60 (33.5%)	31 (36%)	29 (31.2%)
2/3	53 (29.6%)	27 (31.4%)	26 (28%)
**Tumor Pathology**			
Differentiation			
Well	26 (14.7%)	15 (17.4%)	11 (11.8%)
Well-Moderate	19 (10.7%)	10 (11.6%)	9 (9.7%)
Moderate	86 (48.6%)	37 (43%)	49 (52.7%)
Moderate-Poor	29 (16.4%)	15 (17.4%)	14 (15.1%)
Poor	17 (9.6%)	7 (8.1%)	10 (10.8%)
Number of Nodules			
1	147 (82.1%)	82 (95.3%)	65 (69.9%)
2	21 (11.8%)	4 (4.7%)	17 (18.3%)
3	5 (2.8%)	0 (0%)	5 (5.4%)
4+	6 (3.4%)	0 (0%)	6 (6.5%)
2+	32 (18%)	4 (4.7%)	28 (30.1%)
Largest Nodule Size (cm) (mean, 95% CI)	6.26 (5.54–6.98)	5.38 (4.27–6.49)	7.07 (6.15–8)
Aggregate Nodule Size (cm) (mean, 95% CI)	6.74 (5.96, 7.51)	5.43 (4.33, 6.54)	7.93 (6.89, 8.98)
Vascular Invasion			
No	128 (71.9%)	77 (89.5%)	51 (54.8%)
Yes	50 (28.1%)	8 (9.3%)	42 (45.2%)
Capsular Involvement	67 (38.5%)	21 (24.4%)	46 (49.5%)
Margin Status			
R0	163 (91%)	79 (91.9%)	84 (90.3%)
≥R1	16 (9%)	7 (8.1%)	9 (9.7%)
Number of Nodes Examined (mean, 95% CI)	0.33 (0.15–0.51)	0.36 (0.04, 0.67)	0.31 (0.12, 0.5)

Abbreviations: AA, African American; HBV, hepatitis b virus; HCV, hepatitis c virus; AFP, alpha-fetoprotein.

**Table 2 cancers-15-02433-t002:** Transplant Criteria of Patients with HCC.

Variables [*n*, % of Patients Meeting Criteria]	Overall (*n* = 179)	No Recurrence (*n* = 86)	Recurrence (*n* = 93)
**Radiologic Criteria**			
Within Milan	92	55 (59.8%)	37 (40.2%)
Outside Milan but within UCSF	26	7 (26.9%)	19 (73.1%)
Outside UCSF	61	24 (39.3%)	37 (60.7%)
Outside Milan	87	41 (47.1%)	56 (52.9%)
**Pathologic Criteria**			
Within Milan	87	60 (69%)	27 (31%)
Outside Milan but within UCSF	23	8 (34.8%)	15 (65.2%)
Outside UCSF	69	18 (26.1%)	51 (73.9%)
Outside Milan	92	26 (28.3%)	66 (71.7%)

**Table 3 cancers-15-02433-t003:** Univariate and Multivariate Predictors of Recurrence.

Variable	Comparison	Univariate HR (95% CI)	*p* Value	Multivariate HR (95% CI)	*p* Value
Patient Characteristic					
Asian Race	Vs. White Race	0.46 (0.28–0.74)	<0.01	0.86 (0.49, 1.51)	0.60
AFP					
100–999	Vs. ≤20	2.37 (1.38–4.08)	<0.01	2.01 (1.06, 3.80)	0.03
≥1000	Vs. ≤20	2.47 (1.42–4.28)	<0.01	2.02 (1.07, 3.82)	0.03
Radiology					
Aggregate Nodule Size (cm)	Per cm diameter	1.05 (1.01-1.09)	0.02	0.97 (0.85, 1.13)	0.81
Beyond Milan but within UCSF	Vs. within Milan	2.43 (1.39–4.24)	<0.01	1.82 (0.93, 3.53)	0.08
Beyond UCSF	Vs. within Milan	1.72 (1.09–2.71)	0.02	1.36 (0.50. 3.69)	0.54
Pathology					
Fibrosis (Batts-Ludwig Criteria)					
1	vs. 0	0.19 (0.06–0.63)	0.01	0.35 (0.09, 1.31)	0.12
Tumor					
Nodule #					
2+	vs. 1	3.11 (1.98–4.88)	<0.01	2.67 (1.623, 4.391)	<0.01
Largest Nodule Size (cm)	Per cm diameter	1.04 (1.00–1.08)	0.03	0.82 (0.67, 1.01)	0.06
Aggregate Nodule Size (cm)	Per cm aggregate diameter	1.05 (1.02–1.09)	<0.01	1.22 (1.02, 1.47)	0.03
Vascular Invasion	Vs. None	3.57 (2.35–5.40)	<0.01	2.25 (1.30, 3.89)	<0.01
Capsular Involvement	Vs. None	1.81 (1.20–2.73)	<0.01	1.12 (0.67, 1.88)	0.66
Transplant Criteria: Beyond Milan but within UCSF	vs. within Milan	3.20 (1.68–6.08)	<0.01	**	
Transplant Criteria: Beyond UCSF	vs. within Milan	3.44 (2.14–5.53)	<0.01	**	

** Not included in multivariate regression due to co-linearity.

**Table 4 cancers-15-02433-t004:** Multivariate predictors of recurrence used in RESTORE index.

Variable	Hazard Ratio	*p*-Value	RESTORE Points
Pre-Op AFP			
≤20	Ref		**0**
21–99	1.37 (0.79–2.26)	0.35	**1**
≥100	1.78 (1.08–2.92\3)	0.02	**2**
Vascular Invasion			
No	Ref		**0**
Yes	2.77 (1.72–4.44)	<0.01	**3**
Lesion No.			
1 lesion w/in Milan	Ref		**0**
1 lesion outside Milan	1.33 (0.79–2.26)	0.29	**1**
2+ lesions	3.39 (1.93–5.97)	<0.01	**4**

Abbreviations: AFP, alpha-fetoprotein.

## Data Availability

The data presented in the study are available on reasonable request from corresponding author.

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
