# Peer review of "Resected Tumor Outcome and Recurrence (RESTORE) Index for Hepatocellular Carcinoma Recurrence after Resection"

_cancers, 2023, doi:10.3390/cancers15092433_

Round 1

Reviewer 1 Report

This is an interesting paper on an important clinical topic. The issue is not novel, but the easy-to-use implemented score may be helpful in clinical practice.

Comments:

(1) The authors did not analyze the role of the ALBI grade as a predictor of HCC recurrence. However, as the authors know, this grade has been validated and incorporated in well-constructed models of HCC recurrence (i.e., J Hepatol 2018; 69: 1284–1293). Therefore, I suggest the authors include the ALBI grade among the variables potentially predicting HCC recurrence and analyze its association with the recurrence by univariate and, eventually, multivariate analysis.

(2) In Table 2 and the corresponding text, it is not clear which stages of liver fibrosis were considered "1" vs. "0". The authors should clarify this point. It seems they compared the absence of fibrosis with any stage of fibrosis. On the contrary, I suggest comparing fibrosis stage < 2 with > 2.

Reviewer 2 Report

The manuscript is really well done

Despite the retrospective nature of the study, the statistical method is strong and accurate; the risk score seems to be flawless

Many congratulations for this valuable work

Reviewer 3 Report

The authors analyzed post-resection outcome of HCC and developed an index for recurrence.

Comments

1. How do you follow the patients after resection?

2. How do you define the recurrence? 

3. What is LRT in Table 1? loco-regeional therapy? is this different from TACE, RFA, or Y-90?

4.  Some items of the numbers in Table 1 are different from the text. For example, steatosis and inflammation.

5. Any references for Brunt fibrosis, steatosis, and inflammation scores?

6. The authors need to do sensitivity analysis separately for primary HCC and recurrent HCC (non-upfront surgery).

7.  Variable and hazard ratio in lesions No. of Table 3 need to correspond correctly.

8.  Is extended UCSF criteria different from UCSF criteria?

Reviewer 4 Report

The authors have developed a simple scoring system that helps predict recurrence post-resection in patients undergoing curative resection for HCC. The scoring rubric is simple to use and educational for both patient and provider, albeit not particularly novel as the factors already contribute to other scoring/prediction tools.

Can the authors further delineate the patient groups with respect to cirrhosis or fibrosis? The numbers don't add up as the total "n" = 179 but adding these groups together is a number >179. This is important as the presence of cirrhosis is a known independent risk factor for HCC (and potentially HCC recurrence). The authors also need to provide more detail regarding the subgroups of steatosis, as this is different in a pre-cirrhotic liver compared with that finding in the setting of cirrhosis.

Can the authors elaborate on why their observation of lower recurrence post-resection occurred in patients within UCSF criteria versus patients within Milan criteria. This is counterintuitive to the scoring system as total tumor size is a factor for recurrence.

The authors divided the patients into 3 strata based on risk. Can they elaborate on how they determined the cut-offs and the determination of the final high risk cut-off (score >5).

Round 2

Reviewer 1 Report

The Authors have addressed the reviewer's comments.

Reviewer 3 Report

I have no other comments.

Reviewer 4 Report

The authors have addressed the comments that were raised in the prior review.